# Factors Associated with Uptake of Routine Measles-Containing Vaccine Doses among Young Children, Oromia Regional State, Ethiopia, 2021

**DOI:** 10.3390/vaccines12070762

**Published:** 2024-07-11

**Authors:** Abyot Bekele Woyessa, Monica P. Shah, Binyam Moges Azmeraye, Jeff Pan, Leuel Lisanwork, Getnet Yimer, Shu-Hua Wang, J. Pekka Nuorti, Miia Artama, Almea M. Matanock, Qian An, Paulos Samuel, Bekana Tolera, Birhanu Kenate, Abebe Bekele, Tesfaye Deti, Getachew Wako, Amsalu Shiferaw, Yohannes Lakew Tefera, Melkamu Ayalew Kokebie, Tatek Bogale Anbessie, Habtamu Teklie Wubie, Aaron Wallace, Ciara E. Sugerman

**Affiliations:** 1Oromia Regional Health Bureau, Addis Ababa P.O. Box 24341, Ethiopiamuletagalmesa@gmail.com (T.D.); 2Health Sciences Unit, Faculty of Social Sciences, Tampere University, 33100 Tampere, Finland; 3Global Immunization Division, Centers for Disease Control and Prevention, Atlanta, GA 30329, USA; hyy9@cdc.gov (M.P.S.); bwf1@cdc.gov (C.E.S.); 4Global One Health Initiative, Ohio State University, Addis Ababa P.O. Box 1176, Ethiopia; 5College of Medicine, Ohio State University, Columbus, OH 43210, USA; 6Global Immunization Division, CDC-Ethiopia, Addis Ababa P.O. Box 3243, Ethiopia; 7UNICEF, Addis Ababa P.O. Box 1169, Ethiopia; 8Ministry of Health of Ethiopia, Addis Ababa P.O. Box 1234, Ethiopia; 9African Field Epidemiology Network, Addis Ababa P.O. Box 12874, Ethiopia; 10Ethiopian Public Health Institute, Addis Ababa P.O. Box 1242, Ethiopia

**Keywords:** measles, measles-containing vaccine, MCV1, MCV2, second year of life, immunization coverage, dropout rate, barriers, Oromia, Ethiopia

## Abstract

Recommended vaccination at nine months of age with the measles-containing vaccine (MCV1) has been part of Ethiopia’s routine immunization program since 1980. A second dose of MCV (MCV2) was introduced in 2019 for children 15 months of age. We examined MCV1 and MCV2 coverage and the factors associated with measles vaccination status. A cross-sectional household survey was conducted among caregivers of children aged 12–35 months in selected districts of Oromia Region. Measles vaccination status was determined using home-based records, when available, or caregivers’ recall. We analyzed the association between MCV1 and MCV2 vaccination status and household, caregiver, and child factors using logistic regression. The caregivers of 1172 children aged 12–35 months were interviewed and included in the analysis. MCV1 and MCV2 coverage was 71% and 48%, respectively. The dropout rate (DOR) from the first dose of Pentavalent vaccine to MCV1 was 22% and from MCV1 to MCV2 was 46%. Caregivers were more likely to vaccinate their children with MCV if they gave birth at a health facility, believe that their child had received all recommended vaccines, and know the required number of vaccination visits and doses. MCV2 coverage was low, with a high measles dropout rate (DOR). Caregivers with high awareness of MCV and its schedule were more likely to vaccinate their children. Intensified demand generation, defaulter tracking, and vaccine-stock management should be strengthened to improve MCV uptake.

## 1. Introduction

The Expanded Program on Immunization (EPI) was established by the World Health Organization (WHO) in 1974 to control vaccine-preventable diseases and encourage Member States to establish the EPI in their respective health care delivery system [1,2]. Following the WHO recommendations, Ethiopia established the EPI in 1980, providing selected vaccines, including measles-containing vaccine (MCV), for free to children [3]. Vaccines are provided to children in many modalities, including through campaigns, at health posts, at health facilities, and via mobile health clinics. However, after more than four decades of implementation of the EPI, the coverage of full immunization remains very low in the country [4,5]. It is estimated that the proportion of fully vaccinated children 12-23 months old has increased from 14% in 2000 to 39% in 2016 [6,7], and to 43% in 2019, in Ethiopia [8]; however, this is below the WHO’s target goal of 90% coverage for all recommended vaccines by 2030 in every country [9,10]. Despite the availability of effective vaccines, as well as national and substantial government and partner efforts, cases of vaccine-preventable diseases such as measles and polio continue to occur in the country [11]. For instance, between January 2019 and January 2020, an ongoing measles outbreak with a total of 9672 measles cases was reported to the WHO from Ethiopia, of which 5820 (60%) were from Oromia Region, where, nationally, children aged less than five years were the most affected [12]. The mortality rate among children under five years old was 55.2 per 1000 live births in 2019 in Ethiopia [13].

In 2001, Ethiopia adopted the goal of measles elimination for the African Region by the year 2020 and began implementing WHO/UNICEF strategies for accelerating the control of measles. Two of these five key strategies include routine immunization and supplemental immunization. As part of the implementation of the elimination strategies, the WHO recommended all countries to include a second routine dose of MCV (MCV2) in the national vaccination schedules, regardless of the level of MCV1 coverage [14]. In February 2019, the Federal Ministry of Health’s EPI introduced MCV2 for children in the second year of life (at 15 months of age) in Ethiopia into the routine immunization schedule to reduce measles morbidity and mortality and accelerate achieving measles elimination goals [15]. While it has been estimated that over 3.3 million children will receive a second dose of this vaccine annually, Ethiopia has experienced low coverage of the measles vaccine, with the most recent data from WHO/UNICEF estimating 56% and 48% coverage in 2022 for the first and second dose of MCV, respectively [16], and a recent mini demographic and health survey (DHS) in Ethiopia similarly estimating coverage at 59% for the first dose of MCV [8]. The drivers of Ethiopia’s coverage rates are multiple, however, it has been found that maternal knowledge of immunization and vaccine-preventable diseases is a common factor influencing coverage [17,18,19].

The introduction of MCV2 offers not only additional protection against measles, but also provides an opportunity to catch children up on the vaccine doses missed in the first year of life and for the integration of other primary healthcare interventions during the second year of life (2YL), such as vitamin A supplementation, nutritional counseling, growth monitoring and promotion, deworming, and pediatric HIV/AIDS care [20].

Oromia Region is the most populous region in Ethiopia and has the third lowest vaccination coverage, thus contributing to a substantial number of unvaccinated children and subsequent measles outbreaks. Per the Ethiopian mini-DHS of 2019, in Oromia Region, routine immunization coverage for MCV1 was 48.7%. Thus, Oromia Region accounts for about half (~600,000) of the unvaccinated children in Ethiopia [21]. Oromia has also been experiencing frequent measles outbreaks; therefore, achieving high coverage of MCV is essential to prevent the spread of outbreaks and achieve the global measles elimination goals. We conducted this survey to examine measles coverage two years after MCV2 introduction and to identify the factors associated with, and barriers to, caretakers vaccinating their age-eligible children with the measles vaccine in select districts of Oromia Region.

## 2. Materials and Methods

### 2.1. Study Setting

This study was carried out in Oromia Region, Ethiopia (Figure 1). Oromia Region is the most populated region, with over 40 million people, or 37% of the national population. Routine immunization services in the region are provided in 8622 public health facilities, which includes 108 hospitals, 1399 health centers, and 7115 health posts. In a district, a primary health care unit consists of 4–5 health centers, and each health center has 4–5 satellite health posts.

### 2.2. Study Design

A cross-sectional household survey was conducted among caregivers of children aged 12–35 months from February to March 2021 in randomly selected communities in Oromia Region, Ethiopia.

### 2.3. Sample Size and Sample Selection

Of the 337 districts across Oromia Region, 18 districts were randomly selected to be included in the survey after an initial stratification by urban/rural settlement and the MCV1 baseline administrative coverage (high: ≥80%, low: <80%) and recent measles outbreak status (Figure 2). The sample size for the MCV coverage estimation was calculated using the revised WHO Vaccination Coverage Cluster Survey Reference Manual [22]. We assumed an expected measles coverage rate of 30–70%, precision of ±10, intraclass correlation coefficient (ICC) of 0.333, and 10% non-response rate as the parameters to calculate stratum-specific sample size (SS), with a total of six strata (as indicated in Figure 2).
$$\begin{array}{c} \mathrm{SS}=\mathrm{Effective}\mathrm{sample}\mathrm{size}\;[\mathrm{ESS}]\times (1+\mathrm{ICC})\times \mathrm{Inflation}\mathrm{factor}\\ =103\times (1+0.333)\times 100\%/(100\%-10\%)\\ =153\\ \mathrm{Total}\mathrm{sample}\mathrm{size}(\mathrm{across}6\mathrm{strata})=153\times 6=\mathrm{918} \end{array}$$

To enroll the required sample size, we randomly selected two communities, known as kebeles, the lowest administrative unit in Ethiopia, which made the total selected kebeles 36 in the selected districts. Furthermore, within each selected kebele, we randomly selected one village, known as a Gere or a Gote in rural and urban areas, respectively (Figure 2). Assuming 30 interviews for each selected Gere/Gote (15 caregivers with children aged 12 to 23 months and 15 caregivers with children aged 24 to 35 months), we rounded up the minimum targeted sample size to 1080 (540 caregivers of children aged 12–23 months and 540 caregivers of children aged 24–35 months). To select households, we surveyed the first household at the center of the selected village and moved to the right of where we started until the number of caregivers with eligible children to interview was reached. If there were multiple eligible children in the selected household, the caregiver of the youngest child was surveyed.

### 2.4. Data Collection

During the interviews, the caregivers were asked about their demographics and child vaccination status. The household characteristics collected included members involved with vaccine decision making for the child, distance from health facility, residing area, and number of children <5 years of age in the household. The child demographics included age at vaccination, place of birth, sex, and availability of the child’s vaccination card. The caregiver demographics included age, education level, religious denomination, marital status, and their awareness, knowledge, attitudes, and practices surrounding vaccination. Child vaccination status for MCV1 and MCV2, the primary outcome of this study, was based on home-based records (HBR, vaccination card), if available, or from the caregivers’ recall if absent. Questionnaires were developed and adapted from standardized questionnaires to assess the vaccination status and behavior and social drivers of vaccination. The structured questionnaires were pre-tested prior to data collection and pre-programmed for skip patterns and logic checks using Open Data Kit (ODK) software (https://getodk.org/) and administered on tablets. Experienced data collectors were recruited and trained on the study procedures, data collection tools, and interview techniques for three full days prior to fieldwork. Additionally, one trained supervisor was assigned to each team to ensure the data were collected as per the protocol and to review data quality.

### 2.5. Data Analysis

Descriptive analyses were stratified by settlement (urban and rural) and age group (12–23 and 24–35 months). The immunization coverage estimates were stratified by settlement and by age group (12–23 and 24–35 months). We calculated the DOR as the difference between the vaccination coverage of the initial and final doses divided by the coverage of the first dose (i.e., Penta1-Penta3 DOR: [Penta1 − Penta3] ÷ Penta1 × 100%), were DORs > 10% reflect underutilization of immunization services.

We performed bivariate regression between the explanatory and outcome variables using logistic regression among children aged 12–23 months for MCV1 and among children aged 18–35 months for MCV2, including children aged 18–23 months here to increase statistical power in the regression. We grouped continuous explanatory variables such as caregiver’s age, distance to nearest health facility, and waiting time to better understand their relationship with the outcome variables. All variables with *p*-value < 0.15 in the bivariate regression analysis were included in the initial multivariate regression model. We used a backward selection approach for model building, and the final model included all variables with *p*-values < 0.05. Odds ratios (ORs) and 95% confidence intervals (CIs) were reported for all unadjusted and adjusted associations. Adjusted ORs (aOR) with 95% CI and *p*-values < 0.05 in the multivariate regression analysis were used as the cutoff point to determine the factors associated with MCV1 and MCV2 vaccination uptake. Data were cleaned and analyzed in R.

## 3. Results

### 3.1. Household, Caregiver, and Child Characteristics

A total of 1185 caregivers were interviewed across the 18 selected districts and 36 selected kebeles for a 97% response rate. Among those that agreed to participate, 1172 (99%) had a child within the eligible age range for this study (12–23 or 24–35 months) and were included in the analysis. With the exception of the caregivers’ educational attainment, the demographic characteristics were similar for both age groups and are described together (12–35 months combined). Nearly all of the caregivers were the child’s mother (96%), and many (38%) were the head of the household. The caregivers had a median age of 26 years (interquartile range (IQR), 24–30), were predominantly Christian (70%) or Muslim (27%), and spoke Afan Oromo as their primary language, with differences in spoken language in the rural areas compared to the urban areas (90% overall; 100% rural, 80% urban). Most were married (97%) and in a monogamous marriage (99%). The most frequently reported occupations among the caregivers were housewife (61%), followed by farmer (15%). About one-third of the caregivers had no formal education, and this was higher in the rural areas compared to the urban areas (32% overall; 52% rural, 11% urban). Notably, the proportion without formal education was higher among the caregivers of children aged 24–25 months in the rural areas (57.5%) compared to the caregivers of children aged 12–23 months in the rural areas (46.8%). The median number of children < 59 months of age living in the household was one (IQR, 1–2). The majority of households (94%) were within 1 to 5 km of a health facility (Table 1).

Nearly half of the children included in the assessment were female (52%). Most of the children were born at a health facility (79%; 94% in urban areas, 64% in rural areas). Over half of the children had HBRs, and HBR retention was higher in the urban areas compared to the rural areas (58%; 46% rural, 70% urban, Table 1).

### 3.2. Vaccination Coverage

Vaccination coverage for both age groups (12–23 and 24–35 months) exceeded 80% for most antigens and doses and was slightly higher among older (24–35 months) compared to younger (12–23 months) children (Table 2A). However, the coverage of the oral polio vaccine at birth (OPV0), MCV1, and MCV2 was sub-optimal (<80%) (Table 2A and Table 3A). Among children 12–23 months of age, OPV0 coverage was 16% and MCV1 coverage was 71%, with higher MCV1 coverage in the urban areas (81%) compared to the rural (61%) areas (Table 2A). The coverage of MCV2 was 48% among children aged 24–35 months, also with higher coverage in the urban areas (53%) compared to rural (42%) settings (Table 3A).

Based on HBR, the proportion of children who were fully immunized for all recommended infant EPI vaccines in Ethiopia (one dose of BCG, three doses of Pentavalent vaccine, three doses of OPV, three doses of PCV, two doses of Rotavirus vaccine, one dose of IPV, and one dose of measles vaccine) was 67% (53% rural, 77% urban) among children aged 12–23 months (Table 2B) and 66% (49% rural, 74% urban) among children aged 24–35 months (Table 3B). However, including MCV2 in the fully immunized definition classified 35% of children aged 24–35 months (22% rural, 42% urban) as being fully vaccinated (Table 2B).

The Penta1 to Penta3 DOR was 8% (14% rural, 4% urban) among children aged 12–23 months and 3% (7% rural, 1% urban) among those aged 24–35 months (Table 2B). The Penta1 to MCV1 DOR was 22% (35% rural, 12% urban) among children aged 12–23 months (Table 2B) and 15% (25% rural, 9% urban) among those aged 24–35 months (Table 3B). The DOR between the first and second doses of MCV was much higher, at 46% (49% rural, 45% urban) among children aged 24–35 months (Table 3B).

### 3.3. Factors Associated with MCV1 Uptake

Based on backwards selection, the final variables in the multivariate model for MCV1 were as follows: caregiver’s education, caregiver’s age, number of children less than five years old living in the household, caregiver’s who believed the child had received all recommended vaccines, and caregiver’s who reported being turned away from the health facility due to vaccine stockout. Caregivers with higher levels of education were more likely to vaccinate their children with MCV1, and the strength of this association generally increased with each additional level of educational attainment (aOR 2.6; 95% CI 1.5 to 4.6 for primary, aOR 3.2; 95% CI 1.6 to 6.6 for secondary, and aOR 6.8; 95% CI 2.9 to 17.8 for tertiary education, as compared to no formal education). The caregivers aged 27 to 80 years were more likely to vaccinate their children with MCV1, as compared to the caregivers aged 18 to 26 years (aOR 1.7 95% CI 1.0 to 2.7). The children residing in households with multiple children under the age of five years were less likely to be vaccinated with MCV1 (aOR 0.6, 95% CI 0.4 to 1.0 for two vs.one child under five years and aOR 0.4, 95% CI 0.1 to 1.1 for three or four vs. one child under five years). The caregivers who reported that their child had received all recommended vaccines were more likely to have a child that was vaccinated with MCV1 (aOR 7.8, 95% CI 4.9 to 12.7) compared to caregivers who did not believe that their child was fully immunized. The caregivers who knew the correct number of vaccination visits required to complete vaccination services were more likely to vaccinate their child with MCV1 (aOR 2.44, 95% CI 1.02 to 6.87). The caregivers who reported ever being turned away from a health center due to vaccine stockout were less likely to vaccinate their child with MCV1 (aOR 0.4, 95% CI 0.2 to 0.8) (Table 4).

### 3.4. Factors Associated with MCV1 Uptake, Stratified by Rural and Urban Settlement

The factors significantly affecting the uptake of MCV1 among children aged 12–23 months differed in rural and urban settings. In the rural areas, only two factors were associated with MCV1 vaccination, as follows: higher levels of educational attainment for the child’s caregiver (aOR 2.0, 95% CI 1.1 to 3.8 for primary; aOR 4.4, 95% CI 1.6 to 12.8 for secondary, and aOR 3.7, 95% CI 0.7 to 20.7 for tertiary education levels, as compared to no formal education) and the caregivers’ belief that their child had received all recommended vaccines (aOR 13.3, 95% CI 7.5 to 24.5) (Appendix A).

In contrast, in urban settings, several different factors were significantly associated with MCV1 vaccination. The caregivers aged 27 to 80 years were more likely to vaccinate their children with MCV1, as compared to the caregivers aged 18 to 26 years (aOR 2.9, 95% CI 1.3 to 6.9). Children residing in households with multiple children under the age of five years were less likely to be vaccinated with MCV1 (aOR 0.25, 95% CI 0.11 to 0.56 for two vs. one child under five years; and aOR 0.09, 95% CI 0.01 to 0.51 for three or four vs. one child under five years). The caregivers who reported that their child had received all recommended vaccines were more likely to have a child who was vaccinated with MCV1 (aOR 4.55, 95% CI 2.17 to 9.91) compared to the caregivers who did not believe that their child was fully immunized. The caregivers who knew the correct number of vaccinations visits (aOR 4.43, 95% CI 1.26 to 23.1) and named measles as a vaccine-preventable disease (aOR 3.75, 95% CI 1.49 to 9.60) were more likely to have vaccinated their child with MCV1. The caregivers who reported ever being sent home from a health center due to vaccine stockout were less likely to vaccinate their child with MCV1 (aOR 0.33, 95% CI 0.13 to 0.86) (Appendix A).

### 3.5. Factors Associated with MCV2 Uptake

Children who were delivered at or on the way to a health facility were likely to be vaccinated with MCV2, as compared to children who were delivered at home (aOR 2.37; 95% CI 1.30 to 4.47). Similar to MCV1, the caregivers who reported that their child had received all recommended vaccines were more likely to have a child that was vaccinated with MCV2 (aOR 8.29, 95% CI 4.52 to 16.3) compared to the caregivers who did not believe that their child was fully immunized. The caregivers who knew the recommended number of vaccination visits (aOR 3.12, 95% CI 1.87 to 5.36) and doses of childhood measles vaccine (aOR: 1.62, CI: 1.05 to 2.50) were more likely to vaccinate their children with MCV2 (Table 5).

### 3.6. Factors Associated with MCV2 Uptake, Stratified by Rural and Urban Settlement

Factors statistically associated with the utilization of MCV2 were mostly similar in urban and rural settings. The caregivers who believed that their child had received all recommended vaccines and who knew that six vaccination visits were needed to receive all of the EPI vaccines were more likely to vaccinate their children with MCV2 in both rural and urban settings.

In rural areas only, the caregivers who accessed the nearest health facility within 30 min to 1 hour (aOR 11.3, 95% CI 1.73 to 2.27) and within 30 min or less (aOR 9.1, 95% CI 1.6 to 1.7) were more likely to vaccinate their children with MCV2, as compared to the caregivers who accessed the health facility within 1–6 h. (Appendix A).

## 4. Discussion

Two years after the introduction of MCV2 in Ethiopia, we found that coverage for both MCV1 and MCV2 in Oromia Region was too low to attain the measles elimination goals. MCV1 coverage in this study was higher compared with previous DHS estimates [7,8], but our study excluded areas inaccessible due to the ongoing conflict where the coverage for routine immunization, including MCV1, could be lower. The MCV2 coverage (48% among children aged 24–35 months), however, was fairly comparable with the findings from other African countries (around 45%) [23]. In addition, the findings from a community-based survey in Tanzania showed slightly lower, but comparable, MCV2 coverage (44%) [24] and lower coverage as compared with the findings in Ghana (67–82%) [25]. MCV2 coverage might be low because it is relatively newly introduced, and often the first vaccine to be introduced in second year of life, as is the case in Ethiopia. In addition, MCV2 was introduced in February 2019, and the COVID-19 pandemic also likely contributed to the lower coverage rate. Similar to MCV1, the coverage for MCV2 was possibly lower than our estimate, as we excluded the inaccessible areas in active conflict. The recent study conducted in March 2022 by Project Hope in remote and underserved communities in Ethiopia also estimated low MCV1 and MCV2 coverage at 66% and 34%, respectively [26].

The coverage for MCV1 and MCV2 was higher in urban areas compared to rural areas, as found in another recent survey [26]. This might be due to the fact that communities in the urban setting have better access to health care services and information. Based on stratified multivariate analyses, households accessing the nearest health facility in less than a 1-hour walk were more likely to vaccinate their children, as compared to households accessing health facilities within a 1–6-h walk in rural areas. This suggested that households situated far from the nearest health facility in the rural area are less likely to vaccinate their children. Establishing locally tailored vaccination sessions, including outreach through discussion with rural communities, could help the communities residing far from the health facility to vaccinate their children.

Although higher levels of educational attainment and awareness of immunizations were associated with MCV1 and MCV2 vaccination, the overall coverage of MCV2 remained low among the study population. The caregivers who believed that their children received all of the recommended doses of the vaccine and those who knew the required number of visits and measles doses were more likely to vaccinate their children. This highlights that adequate knowledge and information about the second dose of the measles vaccine may not be sufficient to increase vaccination uptake. Another study conducted in multiple African countries also reported insufficient sensitization and awareness generation among parents for low MCV2 coverage [27]. The lack of information was also associated with low MCV2 coverage in Kenya [28]. Therefore, building the health workers interpersonal communication skills to provide key immunization messages to all caregivers at each vaccination session is important for increasing the uptake of MCV1 and MCV2. Increasing parental vaccine decision-making power is an essential factor that should be considered when implementing strategies to improve vaccination uptake. As reported in different studies, face-to-face caregiver and health worker interactions are the most recommended strategy to influence parental decisions to vaccinate their children [29]. On the other hand, our study further uncovered the high vaccination dropout rates between Penta1 and MCV1, and MCV1 and MCV2. The dropout rates were relatively high among the rural areas, as compared with urban dwellers. The MCV1 to MCV2 dropout rate was similar to that from a report from Kenya, which was 46.7% [28]. The higher the vaccination dropout rate could be attributed to the absence or inadequate routine immunization defaulter tracking system in the healthcare facilities. Establishing a locally tailored defaulter identification and tracking systems may have a positive effect on the reduction in vaccination dropout rates.

The caregivers who reportedly experienced being sent home from health facilities due to vaccine stockout were less likely to vaccinate their children with MCV1. This is a missed opportunity for vaccination, as the caregivers might come from far away, and may not return for vaccination again. Vaccine stockout was previously documented as being associated with a missed opportunity for vaccination [30]. The monitoring of vaccines and vaccine supply, maintaining the minimum recommended stock level, and the timely submitting of vaccine requests from the health facility level is important to avoid supply constraints in the health facility settings and to reduce missed vaccination opportunities.

Our survey further revealed that the coverage for OPV0 was low, at 16%, while nearly 80% of children were born in health facilities. They had an opportunity to receive OPV0 at birth, as most of them had contact with a health facility and healthcare providers. This indicates a potential missed opportunity for all vaccines, including MCV1 and MCV2, which might be high. Therefore, service integration in healthcare settings, especially at delivery wards, could minimize potential missed opportunities for vaccination. Further research is needed to understand the magnitude of missed opportunities for vaccination and to identify where and why the children visiting the health facilities, including newborns, are missing the vaccine doses for which they are eligible. Plus, the gaps in birth dose coverage should be explored in advance of the introduction of Hepatitis B birth dose in the future.

This study has some limitations. Conflict-affected areas were purposely excluded from the sampling frame, and the findings of this study might not reflect the trends in those areas. Additionally, because of the unavailability of an updated population census, we could not conduct a weighted analysis to account for any sampling error and infer the findings to the general population. We also used the caregiver’s recall in the absence of HBR to assess the vaccination status of the children. This might lead to an overestimation or underestimation of vaccination coverage. However, the evidence showed that the concordance between HBR and the caregiver’s recall was relatively high in Ethiopia [31]. The discordance between the caregiver’s and HBR was reported as minimal for the measles vaccines [32].

## 5. Conclusions

Two years after MCV2 introduction, coverage remains low, with high measles vaccination dropout rates that illustrate low utilization. The high vaccination dropout rate between MCV1 and MCV2 indicates that the routine immunization defaulter identification, tracking, and follow-up system is weak in healthcare facilities. Caregivers with more awareness of the measles vaccine and its schedule were more likely to vaccinate their children. To improve the uptake of MCV2 in the second year of life and achieve the global measles elimination goals, demand generation, including social mobilization, should be strengthened in the Oromia Region of Ethiopia. In addition, strengthening locally tailored immunization defaulter identification and tracking systems in healthcare facilities is needed to reduce the vaccination dropout rate and increase its utilization. Regular monitoring of vaccines and vaccine supplies and enhancing service integration could have an impact on reducing missed opportunities for vaccination in healthcare settings.

## Figures and Tables

**Figure 1 vaccines-12-00762-f001:**
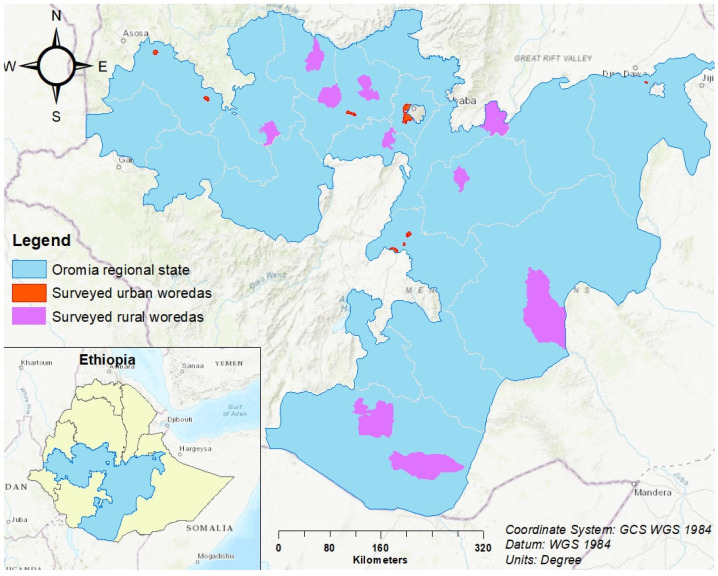
Map showing MCV2 barriers of study surveyed districts (woredas), Oromia region, Ethiopia.

**Figure 2 vaccines-12-00762-f002:**
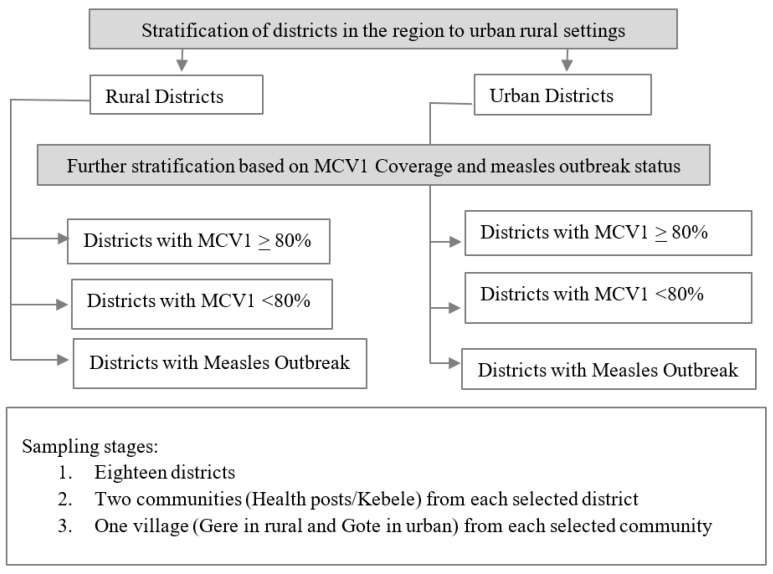
Sample selection flow diagram.

**Table 1 vaccines-12-00762-t001:** Demographic characteristics of households, caregivers, and children surveyed in Oromia Region, Ethiopia.

Characteristic	Total	12–23			24–35		
		All Areas	Rural	Urban	All Areas	Rural	Urban
	N = 1172 n (%)	N = 598 n (%)	N = 299 n (%)	N = 299 n (%)	N = 574 n (%)	N = 287 n (%)	N = 287 n (%)
Relationship to child							
Mother	1124 (95.9)	583 (97.5)	291 (97.3)	292 (97.7)	541 (94.3)	262 (91.3)	279 (97.2)
Father	30 (2.6)	8 (1.3)	4 (1.3)	4 (1.3)	22 (3.8)	19 (6.6)	3 (1.0)
Other relative	18 (1.5)	7 (1.2)	4 (1.3)	3 (1.0)	11 (1.9)	6 (2.1)	5 (1.7)
Caretaker’s age in years, median, IQR	26 (24, 30)	26 (23, 30)	25 (22, 30)	26 (24, 29)	27 (25, 30)	27 (24, 30)	27 (25, 30)
Religion							
Christian	726 (69.5)	373 (69.9)	179 (66.3)	194 (73.5)	353 (69.2)	171 (66.5)	182 (71.9)
Muslim	277 (26.5)	142 (26.6)	75 (27.8)	67 (25.4)	135 (26.5)	70 (27.2)	65 (25.7)
Traditionalist	36 (3.4)	15 (2.8)	12 (4.4)	3 (1.1)	21 (4.1)	15 (5.8)	6 (2.4)
None/Atheist	5 (0.5)	4 (0.7)	4 (1.5)	0 (0.0)	1 (0.2)	1 (0.4)	0 (0.0)
Language							
Afaan Oromo	1054 (89.9)	547 (91.5)	298 (99.7)	249 (83.3)	507 (88.3)	287 (100)	220 (76.7)
Amharic	116 (9.9)	50 (8.4)	1 (0.3)	49 (16.4)	66 (11.5)	0 (0.0)	66 (23.0)
Other	2 (0.2)	1 (0.2)	0 (0.0)	1 (0.3)	1 (0.2)	0 (0.0)	1 (0.3)
Marital status							
Married	1137 (97.0)	583 (97.5)	291 (97.3)	292 (97.7)	554 (96.5)	273 (95.1)	281 (97.9)
Divorced	20 (1.7)	9 (1.5)	3 (1.0)	6 (2.0)	11 (1.9)	6 (2.1)	5 (1.7)
Single	8 (0.7)	3 (0.5)	2 (0.7)	1 (0.3)	5 (0.9)	5 (1.7)	0 (0.0)
Widowed	5 (0.4)	2 (0.3)	2 (0.7)	0 (0.0)	3 (0.5)	2 (0.7)	1 (0.3)
Co-habitation	2 (0.2)	1 (0.2)	1 (0.3)	0 (0.0)	1 (0.2)	1 (0.3)	0 (0.0)
Marriage type							
Monogamous	1145 (99.0)	583 (99.8)	298 (99.7)	285 (100)	562 (98.3)	282 (98.3)	280 (98.2)
Polygamous	11 (1.0)	1 (0.2)	1 (0.3)	0 (0.0)	10 (1.7)	5 (1.7)	5 (1.8)
Caretaker’s Occupation							
Housewife	709 (60.5)	375 (62.7)	184 (61.5)	191 (63.9)	334 (58.2)	158 (55.1)	176 (61.3)
Farmer	163 (13.9)	76 (12.7)	72 (24.1)	4 (1.3)	87 (15.2)	78 (27.2)	9 (3.1)
Professional	145 (12.4)	63 (10.5)	10 (3.3)	53 (17.7)	82 (14.3)	30 (10.5)	52 (18.1)
Self-employed	113 (9.6)	55 (9.2)	19 (6.4)	36 (12.0)	58 (10.1)	13 (4.5)	45 (15.7)
Merchant	18 (1.5)	9 (1.5)	6 (2.0)	3 (1.0)	9 (1.6)	7 (2.4)	2 (0.7)
Student	10 (0.9)	8 (1.3)	3 (1.0)	5 (1.7)	2 (0.3)	0 (0.0)	2 (0.7)
Daily laborer	8 (0.7)	6 (1.0)	4 (1.3)	2 (0.7)	2 (0.3)	1 (0.3)	1 (0.3)
Other	6 (0.5)	6 (1.0)	1 (0.3)	5 (1.7)	0 (0.0)	0 (0.0)	0 (0.0)
Highest level of education completed							
No formal education	372 (31.7)	171 (28.6)	140 (46.8)	31 (10.4)	201 (35.0)	165 (57.5)	36 (12.5)
Primary	409 (34.9)	210 (35.1)	115 (38.5)	95 (31.8)	199 (34.7)	83 (28.9)	116 (40.4)
Secondary	222 (18.9)	121 (20.2)	36 (12.0)	85 (28.4)	101 (17.6)	28 (9.8)	73 (25.4)
Tertiary	169 (14.4)	96 (16.1)	8 (2.7)	88 (29.4)	73 (12.7)	11 (3.8)	62 (21.6)
Caretaker is head of household							
Yes		225 (37.6)	89 (29.8)	136 (45.5)	224 (39.0)	84 (29.3)	140 (48.8)
Head of household’s occupation							
Farmer	388 (33.2)	192 (32.2)	175 (58.7)	17 (5.7)	196 (34.4)	175 (61.4)	21 (7.4)
Self-employed	380 (32.6)	183 (30.7)	48 (16.1)	135 (45.2)	197 (34.6)	48 (16.8)	149 (52.3)
Professional	296 (25.4)	158 (26.5)	39 (13.1)	119 (39.8)	138 (24.2)	45 (15.8)	93 (32.6)
Other	59 (5.1)	40 (6.7)	27 (9.1)	13 (4.3)	19 (3.3)	10 (3.5)	9 (3.2)
Housewife	44 (3.8)	24 (4.0)	9 (3.0)	15 (5.0)	20 (3.5)	7 (2.5)	13 (4.6)
Number of children under 59 months old living in household, median (IQR)	1 (1, 2)	1 (1, 2)	1 (1, 2)	1 (1, 2)	1 (1, 2)	1 (1, 2)	1 (1, 2)
Distance to nearest health facility							
1 to 5 km	1102 (94.2)	555 (93.1)	266 (89.0)	289 (97.3)	547 (95.3)	271 (94.4)	276 (96.2)
6 to 10 km	32 (2.7)	20 (3.4)	13 (4.3)	7 (2.4)	12 (2.1)	2 (0.7)	10 (3.5)
11 to 15 km	20 (1.7)	14 (2.3)	13 (4.3)	1 (0.3)	6 (1.0)	6 (2.1)	0 (0.0)
15 to 30 km	16 (1.4)	7 (1.2)	7 (2.3)	0 (0.0)	9 (1.6)	8 (2.8)	1 (0.3)
Home-based record (HBR) available							
Yes	674 (57.5)	395 (66.1)	169 (56.5)	226 (75.6)	279 (48.6)	99 (34.5)	180 (62.7)
Sex of child							
Male	558 (47.7)	276 (46.2)	130 (43.5)	146 (48.8)	282 (49.2)	127 (44.3)	155 (54.2)
Female	613 (52.3)	322 (53.8)	169 (56.5)	153 (51.2)	291 (50.8)	160 (55.7)	131 (45.8)
Delivery location							
Home	240 (20.5)	115 (19.2)	97 (32.4)	18 (6.0)	125 (21.8)	108 (37.6)	17 (5.9)
On the way to facility	9 (0.8)	6 (1.0)	5 (1.7)	1 (0.3)	3 (0.5)	2 (0.7)	1 (0.3)
Health facility	922 (78.7)	477 (79.8)	197 (65.9)	280 (93.6)	445 (77.7)	177 (61.7)	268 (93.7)

IQR = interquartile range; km = kilometer.

**Table 2 vaccines-12-00762-t002:** Child immunization coverage and indicators based on home-based records (HBR) and caregiver recall among children aged 12–23 months in Oromia Region, Ethiopia, by settlement type.

**A: Child Immunization Coverage Based on Home-Based Records (HBR) If Available and Caregiver Recall, N** **= 598**						
**Vaccine Dose**	**All Areas, N = 598**		**Rural, N = 299**		**Urban, N = 299**	
	**n (%)**	**95% CI**	**n (%)**	**95% CI**	**n (%)**	**95% CI**
BCG	528 (88)	85, 91	258 (86)	82, 90	270 (90)	86, 93
OPV0	96 (16)	13, 19	24 (8.0)	5.3, 12	72 (24)	19, 29
OPV1	544 (91)	88, 93	267 (89)	85, 92	277 (93)	89, 95
OPV2	527 (88)	85, 91	251 (84)	79, 88	276 (92)	89, 95
OPV3	504 (84)	81, 87	236 (79)	74, 83	268 (90)	85, 93
Penta1	544 (91)	88, 93	267 (89)	85, 92	277 (93)	89, 95
Penta2	527 (88)	85, 91	251 (84)	79, 88	276 (92)	89, 95
Penta3	501 (84)	81, 87	233 (78)	73, 82	268 (90)	85, 93
PCV1	545 (91)	88, 93	268 (90)	85, 93	277 (93)	89, 95
PCV2	526 (88	85, 90	250 (84)	79, 88	276 (92)	89, 95
PCV3	501 (84)	81, 87	233 (78)	73, 82	268 (90)	85, 93
Rota1	545 (91)	88, 93	268 (90)	85, 93	277 (93)	89, 95
Rota2	526 (88)	85, 90	250 (84)	79, 88	276 (92)	89, 95
MCV1	423 (71)	67, 74	182 (61)	55, 66	241 (81)	76, 85
IPV ^1^	330 (80)	76, 84	129 (72)	65, 79	201 (86)	81, 90
MCV2 ^2^	100 (50)	43, 57	33 (37)	27, 48	67 (61)	51, 70
**B: Child immunization indicators based on home-based records (HBR) only, N = 395**						
**Vaccine dose**	**All areas, N = 395**		**Rural, N = 169**		**Urban, N = 226**	
	**n (%)**	**95% CI**	**n (%)**	**95% CI**	**n (%)**	**95% CI**
Fully immunized for infant vaccines ^3^	263 (67)	62, 71	89 (53)	45, 60	174 (77)	71, 82
Penta1 to Penta3 DOR ^4^	32 (8.2)	5.7, 11	24 (14)	9.6, 21	8 (3.6)	1.7, 7.1
Penta1 to MCV1 DOR	87 (22)	18, 27	59 (35)	28, 43	28 (12)	8.6, 18

^1^ IPV was not consistently assessed through caregiver recall, therefore, these results reflect mainly HBR data (the denominator for IPV was 412 (178 in rural and 234 in urban). ^2^ MCV2 was not consistently assessed through caregiver recall, therefore, these results reflect mainly HBR data and the coverage estimated among children aged 18–23 months (the denominator for MCV2 was 200 (89 in rural, 111 in urban). ^3^ Fully immunized children in the first year of life as Ethiopia’s EPI criteria is defined as a child receiving 1 dose of BCG, 3 doses of DPT-Hib-HepB (Pentavalent vaccine), 3 doses of OPV, 3 doses of PCV, 2 doses of Rotavirus vaccine, 1 dose of IPV, and 1 dose of measles-containing vaccine. ^4^ DOR denotes dropout rate, CI: Confidence interval.

**Table 3 vaccines-12-00762-t003:** Child immunization coverage and indicators based on home-based records (HBR) and caregiver recall among children aged 24–35 months in Oromia Region, Ethiopia, by settlement type.

**A: Child Immunization Coverage Based on Home-Based Records (HBR) If Available and Caregiver Recall** **, N = 574**						
**Vaccine Dose**	**All areas, N = 574**		**Rural, N = 287**		**Urban, N = 287**	
	**n (%)**	**95% CI**	**n (%)**	**95% CI**	**n (%)**	**95% CI**
BCG	481 (84)	80, 87	224 (78)	73, 83	257 (90)	85, 93
OPV0	70 (12)	9.7, 15	12 (4.2)	2.3, 7.4	58 (20)	16, 25
OPV1	491 (86)	82, 88	232 (81)	76, 85	259 (90)	86, 93
OPV2	486 (85)	81, 87	224 (78)	73, 83	262 (91)	87, 94
OPV3	462 (80)	77, 84	204 (71)	65, 76	258 (90)	86, 93
Penta1	491 (86)	82, 88	231 (80)	75, 85	260 (91)	86, 94
Penta2	481 (84)	80, 87	219 (76)	71, 81	262 (91)	87, 94
Penta3	462 (80)	77, 84	203 (71)	65, 76	259 (90)	86, 93
PCV1	491 (86)	82, 88	231 (80)	75, 85	260 (91)	86, 94
PCV2	480 (84)	80, 87	220 (77)	71, 81	260 (91)	86, 94
PCV3	459 (80)	76, 83	204 (71)	65, 76	255 (89)	84, 92
Rota1	492 (86)	83, 88	233 (81)	76, 85	259 (90)	86, 93
Rota2	479 (83)	80, 86	219 (76)	71, 81	260 (91)	86, 94
MCV1	425 (74)	70, 78	187 (65)	59, 71	238 (83)	78, 87
IPV ^1^	235 (74)	69, 79	76 (59)	50, 68	159 (85)	78, 89
MCV2 ^1^	181 (48)	43, 54	65 (42)	34, 50	116 (53)	46, 60
**B: Child immunization indicators based on home-based records (HBR) only, N = 279**						
**Vaccine dose**	**All areas, N = 279**		**Rural, N = 99**		**Urban, N = 180**	
	**n (%) ^2^**	**95% CI**	**n (%) ^2^**	**95% CI**	**n (%) ^2^**	**95% CI**
Fully immunized for infant vaccines ^2^	183 (66)	60, 71	49 (49)	39, 60	134 (74)	67, 81
Fully immunized for infant vaccines and MCV2 ^3^	97 (35)	29, 41	22 (22)	15, 32	75 (42)	34, 49
Penta1 to Penta3 DOR	9 (3.3)	1.6, 6.4	7 (7.2)	3.2, 15	2 (1.1)	0.20, 4.5
Penta1 to MCV1 DOR	40 (15)	11, 19	24 (25)	17, 35	16 (9.0)	5.4, 15
MCV1 to MCV2 DOR	109 (46)	40, 53	36 (49)	37, 60	73 (45)	37, 53

^1^ IPV and MCV2 were not consistently assessed through caregiver recall, therefore, these results reflect mainly HBR data (the denominator was 316 for IPV and 371 for MCV2). ^2^ Fully immunized children in the first year of life as per Ethiopia’s EPI criteria, defined as a child who received 1 dose of BCG, 3 doses of DPT-Hib-HepB (Pentavalent vaccine), 3 doses of OPV, 3 doses of PCV, 2 doses of Rotavirus vaccine, 1 dose of IPV, and 1 dose of measles-containing vaccine. ^3^ Fully immunized children in the second year of life, defined as a child who received 1 dose of BCG, 3 doses of DPT-Hib-HepB (Pentavalent vaccine), 3 doses of OPV, 3 doses of PCV, 2 doses of Rotavirus vaccine, 1 dose of IPV, and 2 doses of measles-containing vaccine. CI: Confidence interval.

**Table 4 vaccines-12-00762-t004:** Bivariate and multivariate association between caregiver, household, and child demographic characteristics, caregiver’s knowledge, attitude, practice, and awareness factors, and first dose of measles-containing vaccine (MCV1) vaccination status among children aged 12–23 months in Oromia Region, Ethiopia (N = 598).

Characteristic	N	MCV1 = Yes	Bivariate Regression			Multivariate Regression		
			OR	95% CI	*p*-Value	aOR	95% CI	*p*-Value
Settlement					<0.001			
Rural	299	182	Ref					
Urban	299	241	2.67	1.85, 3.88				
Caregiver’shighest level of education completed					<0.001			<0.001
No formal education	171	88	Ref			Ref		
Primary	210	149	2.30	1.51, 3.53		2.59	1.49, 4.59	
Secondary	121	101	4.76	2.75, 8.56		3.21	1.60, 6.63	
Tertiary	96	85	7.29	3.77, 15.3		6.83	2.89, 17.8	
Caregiver’s age in years					0.085			0.039
18 to 26 years	330	225	Ref					
27 to 80 years	264	197	1.37	0.96, 1.97		1.66	1.03, 2.72	
Number of children under 59 months old living in household					<0.001			0.039
One	383	294	Ref			Ref		
Two	188	116	0.49	0.33, 0.71		0.58	0.35, 0.96	
Three or four	26	12	0.26	0.11, 0.58		0.37	0.12, 1.11	
Sex of child					0.75			
Male	276	197	Ref					
Female	322	226	0.94	0.66, 1.34				
Delivery location					<0.001			
Home	115	55	Ref					
At HF or on the way to HF	483	368	3.99	2.29, 5.33				
Caregiver believes that child has received all recommended vaccines					<0.001			<0.001
No/Do not remember	242	108	Ref			Ref		
Yes	355	315	9.77	6.51, 14.9		7.80	4.88, 12.7	
Number of vaccination visits child needs					<0.001			
0–5 visits	449	307	Ref			Ref		
6 visits (correct as per EPI schedule)	76	70	5.40	2.48, 14.2		2.44	1.02, 6.87	0.045
Named measles as a VPD					0.002			
No	96	55	Ref					
Yes	502	368	2.05	1.30, 3.21				
Heard of immunization against measles					0.003			
No	92	53	Ref					
Yes	506	370	2.00	1.26, 3.16				
Number of doses of measles vaccine that child is supposed to receive					<0.001			
Never heard of measles vaccine or do not know number of doses	347	224	Ref					
Heard of measles vaccine—One dose	92	67	1.47	0.89, 2.48				
Heard of measles vaccine—Two doses	151	126	2.77	1.73, 4.56				
Know of a family or community member who had measles					0.17			
No	428	296	Ref					
Yes	170	127	1.32	0.89, 1.98				
In the household, who makes the decision to immunize the child?					<0.001			
Mother or father only (one parent)	133	73	Ref					
Both father and mother	456	345	2.55	1.71, 3.82				
Ever been sent home from health center due to vaccine stockout?					<0.001			0.005
No	514	381	Ref					
Yes	80	40	0.35	0.22, 0.56		0.40	0.21, 0.76	
Type of vaccination services available to your child								
Health facility (fixed)	403	299	Ref		0.005			
Outreach site	13	5	0.22	0.06, 0.67				
Both	179	117	0.66	0.45, 0.96				
Frequency of vaccination availability					0.008			
Every month	303	223	Ref					
Every week	171	127	1.04	0.68, 1.60				
Every day	56	51	3.66	1.54, 10.8				
Walking time to vaccination center					0.33			
From 1 to 6 h	39	25	Ref					
30 min to 1 h	85	56	1.08	0.48, 2.38				
30 min or less	461	333	1.46	0.72, 2.85				
How long do you wait at the vaccination center before the child is vaccinated?								
From 1 to 6 h	129	94	Ref		0.50			
30 min to 1 h	145	98	0.78	0.46, 1.30				
30 min or less	311	226	0.99	0.62, 1.56				

N: Total number of surveyed children, MCV: Measles-containing vaccine, OR: Odds ratio, aOR: Adjusted odds ratio, CI: Confidence interval, HF: Health facility, EPI: Expanded program on vaccination, VPD: Vaccine-preventable disease. All independent variables with *p*-value < 0.15 in the bivariate analysis were added to the initial multivariate regression model.

**Table 5 vaccines-12-00762-t005:** Bivariate and multivariate association between caregiver, household, and child demographic characteristics, caregiver’s knowledge, attitude, practice, and awareness factors, and second dose of measles-containing vaccine (MCV2) vaccination status among children aged 18–35 months in Oromia Region, Ethiopia (n = 572).

Characteristic	N	MCV2 = Yes	Bivariate Regression			Multivariate Regression		
			OR	95% CI	*p*-Value	aOR	95% CI	*p*-Value
Settlement					<0.001			
Rural	246	96	Ref					
Urban	328	184	1.91	1.36, 2.67				
Caregiver’s highest level of education completed					<0.001			
No formal education	155	54	Ref					
Primary	198	93	1.69	1.10, 2.61				
Secondary	119	76	3.31	2.02, 5.48				
Tertiary	102	57	2.37	1.42, 3.97				
Caregiver’s age in years					0.20			
18 to 26 years	280	130	Ref					
27 to 80 years	293	149	1.14	0.82, 1.59				
Number of children under 59 months old living in household					0.018			
One	387	196	Ref					
Two	170	80	0.89	0.62, 1.27				
Three or four	14	2	0.16	0.03, 0.61				
Sex of child					0.15			
Male	275	125	Ref					
Female	299	155	1.24	0.89, 1.72				
Delivery location					0.002			0.005
Home	70	20	Ref			Ref		
At HF or on the way to HF	504	260	2.31	1.37, 4.00		2.37	1.30, 4.47	
Caregiver believes that child has received all recommended vaccines					<0.001			<0.001
No/Do not remember	122	16	Ref			Ref		
Yes	451	263	10.2	5.91, 18.7		8.29	4.52, 16.3	
Number of vaccination visits child needs					<0.001			<0.001
0–5	397	174	Ref			Ref		
6 (correct as per EPI schedule)	108	83	3.96	2.47, 6.53		3.12	1.87, 5.36	
Named measles as a VPD					0.051			
No	82	32	Ref					
Yes	492	248	1.60	1.00, 2.60				
Heard of immunization against measles					<0.001			
No	63	15	Ref					
Yes	511	265	3.47	1.94, 6.57				
Number of doses of measles vaccine that child is supposed to receive					<0.001			0.028
Never heard of measles vaccine or do not know number of doses	293	119	Ref			Ref		
Heard of measles vaccine— One dose	77	32	1.03	0.61, 1.70		0.82	0.45, 1.46	
Heard of measles vaccine— Two doses	198	125	2.47	1.71, 3.59		1.62	1.05, 2.50	
Know of a family or community member who had measles					0.33			
No	385	181	Ref					
Yes	189	99	1.19	0.84, 1.69				
In the household, who makes the decision to immunize the child?					<0.001			
Mother or father only (one parent)	121	42	Ref					
Both father and mother	450	235	2.07	1.37, 3.17				
Ever been sent home from health center due to vaccine stockout?					0.22			
No	495	248	Ref					
Yes	73	30	0.74	0.44, 1.20				
Type of vaccination services available to your child					0.82			
Health facility (fixed)	420	206	Ref					
Outreach site	3	2	2.10	0.20, 45.3				
Both	150	71	0.99	0.68, 1.44				
Frequency of vaccination availability					0.46			
Every month	293	142	Ref					
Every week	157	84	1.19	0.81, 1.76				
Every day	71	41	1.34	0.79, 2.26				
Walking time to vaccination center					0.59			
From 1 to 6 h	47	26	Ref					
30 min to 1 h	83	37	0.72	0.35, 1.47				
30 min or less	430	206	0.74	0.40, 1.35				
How long do you wait at the vaccination center before the child is vaccinated?					0.24			
From 1 to 6 h	118	64	Ref					
30 min to 1 h	145	67	0.68	0.41, 1.10				
30 min or less	305	147	0.72	0.47, 1.11				

N: Total number of surveyed children, MCV: Measles-containing vaccine, OR: Odds ratio, aOR: Adjusted odds ratio, CI: Confidence interval, HF: Health facility, EPI: Expanded program on vaccination, VPD: Vaccine-preventable disease. All independent variables with *p*-value < 0.15 in the bivariate analysis were added to the initial multivariate regression model.

## Data Availability

The datasets used to prepare this report are available at the Oromia Regional Health Bureau and can be obtained from the corresponding author upon reasonable request.

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
