# Peer review of "Factors Associated with Uptake of Routine Measles-Containing Vaccine Doses among Young Children, Oromia Regional State, Ethiopia, 2021"

_vaccines, 2024, doi:10.3390/vaccines12070762_

Round 1

Reviewer 1 Report

Comments and Suggestions for Authors

This important study aims to examine measles coverage two years after MCV2 introduction and to identify factors associated with, and barriers to, caretakers taking their age eligible children for routine MCV2 vaccination in select districts of Oromia Region in Ethiopia. The manuscript, however, lacks some important information in the main text. Here are some points I would like the authors to consider to further highlight the contribution of the study.

1.     Line 57-58, Introduction. “WHO’s target goal of 90% coverage for all 57 recommended vaccines by 2020 in every country”. Where can you find this WHO goal? I can’t find the reference source.

2.     Introduction. Please explain the reason why do you choose MCV as the study object, not other childhood vaccines, such as DTP or HepB?

3.     In Line 115-118, Please add more details on the sample size calculation process and formula.

4.     In the method section, could you please add a sample selection diagram? It will help readers to know the process of sample selection and data collection clearly.

5.     In Table 3B, the total N=397 or 395?

6.     In Table 4B, the total N=297 or 279? Please check all numbers carefully and make sure they are correct.

7.     In the Table 5 and 6, please add what variable you have adjusted for in the multivariate regressions? It is unclear now.

8.     In the discussion section, you should compare your results with other studies, especially with other low-and-middle-income country study.

Author Response

Thank you very much for the constructive comments and suggestions. We attached point by point response here

Reviewer 2 Report

Comments and Suggestions for Authors

This is a very well written and important paper on measles vaccination coverage in Ethiopia. Authors did a really good job describing the results and I have only one minor comment below. Other that that, I was impressed by the manuscript.

The description of the district selection is too vague and seems incomplete. The authors write: (Of the 337 districts across Oromia Region, 18 districts were randomly selected to be 112 included in the survey after an initial stratification on urban/rural settlement (nine rural 113 and nine urban districts). It is unlikely to get 9/9 split if it was truly random, so perhaps the 9/9 split was by design?

Author Response

(The authors gave the same response as above.)

Reviewer 3 Report

Comments and Suggestions for Authors

Woyessa and colleagues provide an excellent report on a cross-sectional household survey conducted among caregivers of children aged 12-35 months in selected districts of Oromia region in Ethiopia to determine factors influencing the uptake of measles-containing vaccine. I found no issues with the findings and conclusions of this report, and it can be published in its current form. 

Author Response

(The authors gave the same response as above.)

Reviewer 4 Report

Comments and Suggestions for Authors

Refer to Editor(s) for comments.

Author Response

(The authors gave the same response as above.)

Reviewer 5 Report

Comments and Suggestions for Authors

Review for ‘Factors associated with update of routine second dose of measles-containing vaccine among young children, Oromia, Ethiopia, 2021’

This article summarizes a cross-sectional household survey conducted among caregivers of children 12-35 months of age in selected districts of Oromia, Ethiopia, with the objective of better understanding factors related to receipt of vaccines, in general, and MCV specifically.  The article can be improved by:

Comments

·         Methods:  The sample size calculations summarized in section 2.3 state that 2,754 caregivers were targeted (153 caregivers from each of the 18 districts.)  Only 1,185 were interviewed.  The authors should comment on the implications of this difference.

·         Methods:  Information should be provided on who collected the data, including whether they were trained. 

·         Methods:  Information is required regarding the specific questions used in the survey form For example: Was the survey form pre-tested?  Did the form use pre-defined questions, such as from coverage surveys?

·         Methods:  Please provide information regarding the accuracy of the data collected. For example:  Was the accuracy of the collected data assessed?  Were photographs of HBR taken to estimate data accuracy?  Did data accuracy vary by whether the HBR was an improvised or standardized vaccination card?

·         Methods/Results:  The descriptive data tables present pieces of the information based on differing age breakdowns of the children:  12-35 months, 12-23 months, 18-35 months, and 24-35 months.  However, the methods state that descriptive results are presented for 12-23 and 24-35 month-old children.  The methods state that logistic regression was performed for children 12-35 months and 18-35 months.  These differing age cutoffs are confusing for the reader, and it is hard to understand how the descriptive data fits with the logistic regression data.  Overall, the results need to be better organized and presented more clearly, respecting the age cutoffs throughout the article.

Additional Comments/Edits:

·         Title:  The title focuses on MCV2, but the results presented are broader, including MCV1 and other vaccinations.  Hence, the title should be changed to reflect the key focus of the article of the article itself changed to fit with the title.

·         Abstract:  ‘DOR’ should be written out as this is the first use of the abbreviation.

·         Objectives: lines 92-95:  The objective of the article is stated as assessing reasons for ‘taking’ children for MCV2 vaccination.  However, the article covers a wider range of issues, such as stock outs and defaulter tracing at the health facilities.   Hence, the main objective should be revised to reflect the content of the article, which focuses more on ‘receipt’ of vaccines rather than caretakers ‘taking’ their children for vaccination.

·         Methods:  Lines 121 to 124 are unclear.  What is a ‘Gere’ and a ‘Gote’?

·         Results:  Table 3A provides information for children aged 12-23 months and excludes MCV2, with MCV2 stated as the overall focus of the article.  Please include MCV2, which will provide the reader with an understanding of MCV2 receipt among those 15-23 months of age. 

·         Discussion:  Another reasonable explanation for the differing admin coverage versus survey coverage is inaccurate administrative coverage figures.  Please include this point.

·         Discussion:  Have logbooks at health facilities been updated to allow health facility staff to record MCV2?  It can take years in some countries for this change to occur, and it is reasonable to assume that health facility staff’s administration of MCV2 would be influenced by being able to record such information in logbooks.

·         Lines 329-331: This wording implies that outreach is not being performed.  Is this really the situation? 

Comments on the Quality of English Language

·         Minor editing of the English is required.  The article would benefit from proofreading to reduce lengthy wording of some sentences and requires some minor edits (ie. Line 368 the word ‘the’ needs be inserted before the word ‘magnitude’. )

Author Response

(The authors gave the same response as above.)

Round 2

Reviewer 5 Report

Comments and Suggestions for Authors

No further comments nor suggestions.  Thank you for your revisions.